# IFN-γ-Preconditioned Human Gingival-Derived Mesenchymal Stromal Cells Inhibit Plasmacytoid Dendritic Cells via Adenosine

**DOI:** 10.3390/biom14060658

**Published:** 2024-06-04

**Authors:** William de Jesús Ríos-Ríos, Sorely Adelina Sosa-Luis, Alexia Almaraz-Arreortua, Patricia Vargas-Benitez, Héctor Ulises Bernardino-Hernández, Jaime Vargas-Arzola, Luis Alberto Hernández-Osorio, María de los Ángeles Romero-Tlalolini, Sergio Roberto Aguilar-Ruiz, Honorio Torres-Aguilar

**Affiliations:** 1Basic and Clinical Immunology Research Department, Faculty of Biochemical Sciences, Universidad Autónoma “Benito Juárez” de Oaxaca (UABJO), Av. Universidad s/n. Cinco Señores, Oaxaca 68120, Mexico; qfbrioswilliam@hotmail.com (W.d.J.R.-R.); minna.leluiso22@gmail.com (S.A.S.-L.); alexiaarreortua1@gmail.com (A.A.-A.); hbernardino@yahoo.com (H.U.B.-H.); vajcquabjo@hotmail.com (J.V.-A.); luisheol@hotmail.com (L.A.H.-O.); 2Dirección General de Asuntos Académicos, Coordinación General de Investigación, Universidad Regional del Sureste, Oaxaca 68150, Mexico; cginvestigacion@urse.edu.mx; 3CONACYT-UABJO, Faculty of Medicine and Surgery, Ex Hacienda de Aguilera S/N, Sur, Oaxaca 68020, Mexico; romerotlalolini@gmail.com; 4Faculty of Medicine and Surgery, Ex Hacienda de Aguilera S/N, Sur, Oaxaca 68020, Mexico; sar_cinvestav@hotmail.com

**Keywords:** human gingival mesenchymal stromal cells, IFN-γ-preconditioning, plasmacytoid dendendritic cells, adenosinergic pathway, CD39, CD73, adenosine

## Abstract

Plasmacytoid dendritic cells (pDCs) are vital players in antiviral immune responses because of their high levels of IFN-α secretion. However, this attribute has also implicated them as critical factors behind the immunopathogenesis of inflammatory diseases, and no currently available therapy can efficiently inhibit pDCs’ aberrant activation. Mesenchymal stromal cells (MSCs) possess stromal immunomodulatory functionality, regulating immune cell activation through several mechanisms, including the adenosinergic (CD39/CD73/adenosine) pathway. The IFN-γ preconditioning of bone marrow MSCs improves their inhibitory properties for therapy applications; however, isolating human gingival tissue-derived MSCs (hGMSCs) is more accessible. These cells have shown better immunomodulatory effects, yet the outcome of IFN-γ preconditioning and its impact on the adenosinergic pathway has not been evaluated. This study first validated the immunoregulatory properties of primary-cultured hGMSCs, and the results showed that IFN-γ preconditioning strengthens CD39/CD73 coexpression, adenosine production, and the regulatory properties of hGMSC, which were confirmed by describing for the first time their ability to reduce pDC activation and their IFN-α secretion and to increase the frequency of CD73+ pDC. In addition, when CD73′s enzymatic activity was neutralized in hGMSCs, adenosine production and the IFN-γ preconditioning effect were restrained. This evidence might be applied to design hGMSCs- and adenosine-based immunotherapeutic strategies for treating inflammatory disorders that are associated with pDC overactivation.

## 1. Introduction

Mesenchymal stromal cells (MSCs) constitute a heterogeneous population of no hematopoietic mesenchymal tissue-derived cells with multipotent differentiation capability. In addition, MSCs have shown extraordinary skills in regulating the activation and functionality of some immune cells [1,2]. MSCs are endowed with cell–cell contact and secreted mechanisms, giving them immunoregulatory properties, a highlight among which is extracellular adenosine generation through the enzymatic activity of CD39 and CD73 [3,4,5]. CD39 and CD73 work in tandem to catalyze the surrounding ATP—CD39 hydrolyzes ATP to AMP; CD73 hydrolyzes AMP and produces extracellular adenosine with multiple inhibitory effects [6]. Thus, CD39, CD73, and adenosine shape “the adenosinergic pathway” and change it from an ATP-proinflammatory milieu to an anti-inflammatory environment [7]. Nevertheless, CD39/CD73 enzymatic activity, which is coordinated in tandem, can occur even when the expression of these ectoenzymes coincide in close proximity on different adjacent cells, generating a “purinergic halo” in their local environment and affecting neighboring cells [8]. Adenosine receptors are expressed on practically all immune cells; their recognition modulates inflammatory responses [9] and the immunoregulatory pathway, as well as the “purinergic halo”, which have been widely described in regulatory T cells (Tregs) and tolerogenic dendritic cells (tDC) as one crucial mechanism for maintaining peripheral immune tolerance [10,11].

MSCs have been isolated from various tissues. Bone-marrow-derived MSCs are the most widely investigated and is considered the “gold standard” for MSC research. Additionally, adipose tissue and the umbilical cordon have also been deeply investigated. Human gingival tissue-derived MSCs (hGMSCs) have attracted special attention in recent years because, compared to other tissue-derived MSCs, they are abundant and easy to obtain through minimally invasive cell isolation techniques. In addition, hGMSCs have shown similar or superior immunoregulatory properties [12,13,14].

The preconditioning of MSCs ex vivo with inflammatory stimuli such as IFN-γ has improved their immunoregulatory properties by enhancing different mechanisms, including the adenosinergic pathway [15,16,17]. However, concerning hGMSCs, limited data exist currently explaining the role of IFN-γ in regulating their immunoregulatory capabilities.

Likewise, although the expression of both ectoenzymes of the purinergic pathway (CD39 and CD73) and adenosine generation from ATP have been described in these cells [13], the effect of IFN-γ preconditioning on their immunoregulatory properties has not been explored. 

Plasmacytoid dendritic cells (pDCs) are immune cells that specialize in developing antiviral immune responses, owing to their optimized capability to produce high IFN-α levels [18]. However, their pivotal function for IFN-α secretion has also placed pDCs as an essential cellular immunomodulator behind the immunopathogenesis of various autoimmune diseases, such as systemic lupus erythematosus (SLE) [19] and Sjögren’s syndrome (SS) [20]. 

Several studies have evidenced the effect provided by the adenosinergic pathway on the immunoregulatory properties of MSCs. However, research about hGMSCs is limited. This study aimed to analyze the impact of the IFN-γ preconditioning on hGMSCs’ regulatory properties, elucidating the contribution of the adenosinergic (CD39/CD73/adenosine) pathway to pDC activation. 

## 2. Materials and Methods

### 2.1. Human Samples

Ethical approval for this study was obtained from the Universidad Regional del Sureste’s (URSE) Bioethics Committee, with the ethical approval code URSE/CEI/2023/026.

Human gingival tissues were obtained from healthy volunteers under informed consent. Gingival tissue was extracted from patients undergoing third molar surgery with no history of periodontal disease at the School of Odontology of the Universidad del Sureste, Oaxaca, México.

Peripheral blood samples were obtained from healthy donors enrolled under informed consent at the University Clinical Laboratory of the Biochemical Sciences Faculty from the Universidad Autónoma “Benito Juárez” de Oaxaca (UABJO).

### 2.2. Isolation of hGMSCs 

An explant method was carried out: the obtained tissues were thoroughly washed with PBS 1X and dissected into 1 to 3 mm pieces. Each explant was resuspended in culture medium using Dulbecco’s modified Eagle’s medium, DEMEM (supplemented with 2% antibiotic/antimycotic and 10% FBS), followed by vortexing for 2 min, after which the tissues were collected, placed in 6-well plates, and cultured for 30 min at 37 °C/5% CO_2_ for favoring initial adherence. Furthermore, a culture medium derived from the vortexing process was added to each well. After 24 h of incubation, the medium was replaced by 1 mL of fresh culture medium, and the tissue was incubated for 10–13 days. Then, when optimal cell migration was observed (Figure 1), the tissues were removed, and the medium was replaced with a fresh culture medium. Subsequently, the culture medium changes were made every 4 days until 70% of cell confluence was reached. Then, the cells were subcultured at 80% confluency using 0.25% Trypsin-EDTA and were used between the 4 and 7 passages for the following experiments. A colony-forming unit efficiency (CFU-F) test was performed: isolated hGMSCs were seeded at 1000 cells per well and cultured at 37 °C/5% CO_2_ for 14 days. The culture medium was changed twice per week. After culture, the cells were washed, fixed with 4% formaldehyde, and stained with 1% violet crystal. The morphological features and CFU-F assay analysis were examined using inverted and phase-contrast microscopy.

### 2.3. In Vitro Differentiation of hGMSCs

For the functionality assessment, multipotent differentiation into osteoblasts, chondrocytes, or adipocytes was performed in specific culture media, with cells at passage 4. For hGMSC differentiation into osteoblasts, 2.5 × 10^3^ cells were cultured with an osteogenic differentiation medium (StemMACS OsteoDiff Media (Miltenyi Biotec, Bergisch Gladbach, Germany)) for 2 weeks. The culture medium was replaced twice per week. Osteogenic differentiation was determined through calcium and mineralized matrix deposition, which was detected with Alizarin Red Oil staining. For chondrogenic differentiation, 5.0 × 10^3^ cells were cultured for three weeks in a chondrogenic differentiation medium (StemMACS ChondroDiff Media (Miltenyi Biotec, Bergisch Gladbach, Germany)). The medium was replaced every 4 days. Chondrocyte differentiation was determined using Safranin O staining. For adipocyte differentiation, 2.5 × 10^3^ cells were cultured with adipogenic medium (StemMACS AdipoDiff Media (Miltenyi Biotec, Bergisch Gladbach, Germany)) for 3 weeks, and the medium was replaced twice per week. Adipocyte induction was determined by the lipid formation of droplets, detected with Oil-Red O staining. All condition cultures were performed in 6-well plates at 37 °C/5% CO_2_.

### 2.4. Peripheral Blood Mononuclear Cells (PBMCs) and pDCs Isolation

First, differential centrifugation was performed to eliminate platelets from the whole blood. Then, the blood samples were diluted with Dulbecco’s phosphate-buffered saline 1× (DPBS, ratio 1:2). PBMCs were obtained by the Ficoll density gradient (Lymphoprep TM STEMCELL Technologies). The PBMCs were washed with DPBS 1× and resuspended in RPMI 1640 culture medium (Corning, Manassas, VA, USA) supplemented with 10% heat-inactivated autologous plasma, 2.0 mM L-glutamine, 1.0 mM sodium pyruvate, 0.1 mM nonessential amino acids, 100 U/mL penicillin, 100 mg/mL streptomycin, and 50 nM 2β-mercaptoethanol. Following the manufacturer’s protocol, pDCs were isolated by double-magnetic cell sorting (Anti-BDCA-4/Neuropilin-1 MicroBead Kit, Miltenyi Biotec, Bergisch Gladbach) from PBMCs. Finally, the pDCs were resuspended in the supplemented RPMI 1640 culture medium. 

### 2.5. Flow Cytometry

For hGMSC characterization, an MSC phenotype KIT was used (Miltenyi Biotec, Bergisch Gladbach, Germany). Cells at passage 4 were harvested with 0.25% trypsin-EDTA, washed, and then resuspended in FACS stain buffer (PBS 1×, FBS 10%, sodium azine 0.1%). For pDCs and PBMCs, the following antibodies were used: anti-BDCA-2/FITC (clone AC144), anti-CD4/VioBlue (clone REA623), anti-CD33/APC (clone REA775), anti-CD39/APC (clone REA739), and anti-CD40/PE (clone HB14), all from Miltenyi Biotec; anti-CD123/Alexa Fluor 700 (clone 6H6), and anti-CD73/Pacific Blue (clone AD2), from BioLegend; anti-CD83/FITC (clone HB15e). A fluorescence minus one (FMO) control was used in all experiments to set thresholds for positive and negative stained cells. 

Cells were incubated in FACS buffer for 15 min with the corresponding antibodies or isotype controls at 4 °C in the dark for extracellular staining. For intracellular staining, the hGMSCs were fixed with 4% formaldehyde for 30 min and permeabilized with Triton X-100 (0.2%) for 1 h at room temperature. The cells were washed with FACS buffer and stained for the corresponding antibodies. Then, 7-AAD staining (BD Pharmingen) was used to exclude the dead cells. The MACSQuant Analyzer 10 cytometer (Miltenyi Biotec, Bergisch Gladbach, Germany) at the National Laboratory of Cytometry (LABNACIT-UNAM-UABJO-UACH) was used for data acquisition, and FlowJo version 10 software was used for data analysis (BD Biosciences, New York, NY, USA).

### 2.6. IFN-γ Preconditioning of hGMSCs (hGMSC-γ)

The 5 × 10^4^ cells at passages 4–7 were allowed to adhere overnight in DMEM culture medium at 37 °C/5% CO_2_. Next, the medium was replaced with RPMI 1640 culture medium supplemented with IFN-γ (500 or 1000 IU/mL), and the cells were incubated for 24 or 48 h. After incubation, the cells were harvested and set aside for further experiments. 

After IFN-γ preconditioning, both stimulated and not-stimulated hGMSCs were analyzed for extracellular adenosine production; briefly, the cells were incubated with a culture medium supplemented with ATP 100 µM for 1 h. Following the kit’s protocol, the supernatants were collected for adenosine detection (Cell Biolabs, Inc., San Diego, CA, USA). The adenosine was detected by fluorometry (Synergy™, HTX Multimode Reader).

### 2.7. Validation of hGMSCs’ Regulatory Effect on PBMCs Activation

The PBMCs were cocultured with hGMSCs during their activation. A mixed leukocyte reaction (MLR) test was performed for the proliferation assay. To this end, purified PBMCs from 2 different donors were cultured in supplemented RPMI 1640 culture medium for 2.5 h to allow the separation of non-adherent mononuclear cells (mainly peripheral blood lymphocytes, named PBLs) and adherent mononuclear cells (primarily antigen-presenting cells (APC), like monocytes) at 37 °C/5% CO_2_. After incubation, the PBLs were labeled with 0.5 µM CFSE (Sigma, Kawasaki-shi, Japan) and then cultured in 96-well plates for MLR testing with allogenic APC cells, in the absence or the presence of hGMSCs or hGMSC-γ, at a 1:3:0.3 ratio (PBL:CPA:hGMSCs) for 5 days. Proliferation was estimated using the CFSE dilution method by flow cytometry.

For the activation and maturation assay, the PBMCs were activated with LPS (0111:B4, Sigma) as follows: 2 × 10^5^ cells were stimulated with LPS (1 µg/mL) for 24 h in the presence or absence of hGMSCs or hGMSC-γ at a 1:10 ratio, respectively. After incubation, the PBMCs were harvested for phenotype analysis. For evaluating the effect of hGMSC-derived soluble factors, the same LPS stimulation protocol was used in the presence of conditioned medium (CM), obtained as described previously.

### 2.8. Evaluation of hGMSCs’ Regulatory Effect on pDC Activation and Functionality 

hGMSCs were cocultured with pDCs during their activation. 2 × 10^4^ pDCs were cultured in 96-well plates for 24 h at 37 °C/5% CO_2_ in RPMI 1640 culture medium supplemented with R848 at 2.5 µg/mL (Invivogen, San Diego, CA, USA) plus IL-3 (10 ng/mL), in the absence or presence of hGMSCs or hGMSC-γ, at a 1:10 ratio (hGMSC:pDC). After incubation, the cells were collected for phenotypic analyses. The supernatants were collected and frozen at −86 °C until IFN-α quantification by ELISA (Invitrogen ThermoFisher, Vienna, Austria). 

The corresponding hGMSC-γ cells were previously treated with the neutralizing antibody anti-CD73 (10 µg/mL) (clone 7G2, Abcam, Waltham, Boston) for 1.5 h before coculturing with pDCs.

### 2.9. Statistical Analysis

Prism (GraphPad Software, LLC., Boston, MA, USA, 10.2.3.403) was used to analyze and plot the data. Statistical analysis was carried out using unpaired non-parametric tests. Each specific statistical test is displayed in the image legends. When quantitative graphics are displayed, these are presented as the median and interquartile range.

## 3. Results

### 3.1. The Human Gingival Tissue-Isolated MSCs Fulfilled the International Society for Cell and Gene Therapy (ISCT) Criteria for MSC Characterization

After the primary culture of the gingival tissue, elongated adherent cells appeared close to the tissue after around 5–7 days (Figure 1A: I). Following tissue removal, the cells expanded to form colonies and were confluent for about 20–25 days (passage 0) (Figure 1A: II and III). The morphological features comprised elongated, flat spindle cells with fibroblast-like morphology at passage 3 (Figure 1A: IV). Moreover, the cells formed CFU-F colonies at low cellular density (Figure 1A: V). Phenotype characterization revealed that the hGMSCs presented high surface expression of CD90, CD105, and CD73 and were negative for HLA-DR and hematopoietic lineage markers (CD34, CD45, and CD19) (Figure 1B). Likewise, the obtained hGMSCs were efficiently differentiated into osteoclast, adipocyte, and chondrocyte groups, as revealed by extracellular mineralized calcium deposition, the formation of lipids drops in the cytoplasm, and extracellular matrix deposition, respectively (Figure 1C). These results demonstrated the feasibility of isolating genuine MSCs from gingival tissue, thereby meeting the ISCT criteria for their characterization. 

### 3.2. hGMSCs Constitutively Co-Express CD73 and CD39 

The analysis of CD73 by fluorescence microscopy (Figure 2A) and flow cytometry (Figure 2B) displayed a high frequency (>99%) and broad localization of CD73 on hGMSCs. The analysis of CD39 and CD73 on hGMSCs by flow cytometry revealed that these cells co-express both molecules on their surface (Figure 2B). This coexpression varied depending on the donor, ranging from 7.7% to 36%. Because CD73 expression is high and constitutive in these cells, the CD39/CD73 variable coexpression relied on CD39, since the frequency of CD39+ hGMSCs showed a wide variability among donors (range = 12–47%). The comparison with the intracellular expression of CD39 revealed no significant differences (Figure 2).

### 3.3. IFN-γ Stimulation Increases CD39 Expression, CD39/CD73 Double-Positive Cell Frequency and Adenosine Production in hGSMCs

hGSMCs were stimulated with two doses of IFN-γ, as described in the Materials and Methods section, to evaluate its effect on each member of the purinergic pathway. The analysis revealed that IFN-γ induces a significant upregulation in the frequency of CD39+ cells (unstimulated = 16.4 ± 5.08; 500 IU/mL = 22.08 ± 9; 1000 IU/mL = 31.6 ± 8.3) and expression (median fluorescence intensity = MFI; *p*-values = 0.0029 for unstimulated vs 1000 IU/mL, and 0.0406 for unstimulated vs 500 IU/mL) in a dose-dependent manner (Figure 3A). However, the frequency of CD73+ cells was slightly reduced, and the CD73 expression was not affected on hGMSCs (Figure 3B). Interestingly, the frequency of double-positive (CD39+CD73+ cells; unstimulated= 20.9 ± 6.8; 500 IU/mL= 25.7 ± 5.31; 100 IU/mL= 31.8 ± 5.1) cells (Figure 3C) and adenosine production (Figure 3D) were significantly increased in the stimulated cells, this being more significant for IFN-γ 1000 IU/mL than 500 IU/mL (*p* = 0.047 for CD39+CD73+ frequency; *p* = 0.015 for adenosine production), reliant upon CD39 expression. 

### 3.4. Primary Cultured hGMSCs Possess Modulatory Effects on PBMC Activation, and IFN-γ-Preconditioning Increases Their Regulatory Properties

The validation process to verify whether primary cultured hGMSCs were endowed with immunoregulatory properties and to evaluate the effect of IFN-γ preconditioning (hGMSC-γ) was conducted on peripheral blood lymphocytes (PBLs) with proliferation challenged in a mixed leukocyte reaction (MLR), as described in the Materials and Methods section. In addition, as a purinergic pathway member with upregulation that has been described as a surface marker associated with peripheral blood human Tregs [21], CD39 expression was additionally analyzed in the CD4+ PBLs during this procedure. The results showed that hGMSC-γ produced around a two-fold inhibition of PBL proliferation compared to hGMSCs (54.12% ± 5 and 26.4% ± 4.7 of inhibition, respectively) (Figure 4A,B). Likewise, although both hGMSCs and hGMSC-γ induced significantly higher CD39 expression on CD4+ PBL, hGMSC-γ caused a significantly superior effect (Figure 4C,D). 

In addition, the immunomodulatory hGMSC capacity and the IFN-γ preconditioning effect were evaluated on the activation (CD40) and maturation (CD83) of LPS-stimulated PBMC. The results revealed that both CD40+ and CD83+ cell frequency, as well as CD40 expression (MFI), were significantly reduced in the presence of both hGMSCs and hGMSC-γ, but the effect was more significant in the presence of hGMSC-γ (Figure 5). 

### 3.5. hGMSC-γ Inhibit pDC Activation and IFN-α Secretion via Adenosine 

Once the immunoregulatory capabilities of primary cultured hGMSCs and the increase of this property induced by their IFN-γ preconditioning had been validated, the effects were analyzed on activation (CD40) and the IFN-α secretion of human pDCs. After pDC isolation from peripheral blood by a double-positive selection process with magnetic cell sorting, as described in the Materials and Methods section, their viability (7AAD negative) and purity (BDCA2+CD123+CD4+CD33-) for functional assays were verified, obtaining percentages of 85.9% ± 7.23 (Figure 6A). 

The pDCs were activated with R848 (selective activating ligand for TLR7), as well as in the presence of hGMSCs or hGMSC-γ, and, due to hGMSC-γ, had showed increased expression of the purinergic pathway members (CD39, CD73, and adenosine); then, because CD73 is the last ectonucleotidase to adenosine production, a fourth pDC activation condition was induced by including hGMSC-γ cells that were previously treated with an anti-CD73 neutralizing antibody (Figure 6). The neutralizing effect of CD73 activity was verified by evaluating adenosine production, confirming an increase in hGMSC-γ compared to hGMSCs and a significant reduction in hGMSC-γ when these cells were pretreated with the neutralizing anti-CD73 antibody (Figure 6B). 

The analysis of pDC activation (CD40) and IFN-α revealed that both hGMSCs and hGMSC-γ significantly inhibit CD40+ frequency and expression in pDC, as well as IFN-α secretion, even when CD73 activity had been neutralized. Strikingly, the inhibitory effect was more significant on CD40 expression (MFI) and IFN-α secretion when the pDCs were activated in the presence of hGMSC-γ, and the effect was reversed in the absence of CD73 activity (Figure 6C,D). 

### 3.6. hGMSC-γ Increase the Frequency of CD73+ pDC

CD39 and CD73 catalyze the extracellular ATP into adenosine. However, their synchronized enzymatic activity can occur even if both ectoenzymes occur in different but adjacent cells. Hence, aiming to analyze the influence of hGMSCs on the expression of the purinergic halo, the expression of these molecules was also examined in pDCs, following the culture conditions as previously described. 

The results showed that pDC activation significantly upregulated CD39 expression compared to unstimulated cells. The presence of hGMSCs did not affect the frequency of CD39+ pDC compared to activated pDC. hGMSC-γ significantly increased CD39+ frequency and expression but showed no changes when the CD73 activity was neutralized (Figure 7A(upper panel),B). 

Conversely, the CD73+ frequency of pDC and CD73 expression also increased after pDC activation. Likewise, the CD73+ pDC frequency and expression were significantly increased by both hGMSCs and hGMSC-γ, but this effect was more significant in the presence of hGMSC-γ; interestingly, the neutralization of CD73 activity reduced the frequency of CD73+ pDC (Figure 7A(lower panel),C).

## 4. Discussion

MSCs derived from several tissues have been tested widely for their immunoregulatory roles. Bone-marrow-derived MSCs (BMMSCs) are the most extensively investigated types. Nevertheless, hGMSCs possess advantages over other tissue-derived MSCs (including BMMSCs). Such a predominance is due to their accessibility through non-invasive methods and their better yield, stability, and immunoregulatory effects [12,13,14]. Hence, hGMSCs have recently emerged as a potential source for obtaining suitable cells with potential immunoregulatory properties. Therefore, this research initially demonstrated the feasibility of isolating genuine MSCs through an explant method, while meeting the ISCT criteria for these cells’ phenotypic and functional characterization (Figure 1) [21]. 

Several studies have elucidated that MSCs can use multiple mechanisms to mediate their immunoregulatory functions, such as cell-cell contact and secreted factors [4,5]. The purinergic pathway (CD39, CD73, and adenosine) is spotlighted due to its involvement in modulating inflammatory environments [22]. This study showed that hGMSCs displayed constitutive but differential CD39 and CD73 expression. On the one hand, as a defining MSC marker, CD73 was homogeneously expressed; on the other hand, CD39 showed low surface expression levels (Figure 2) with an intracellular presence, suggesting that CD39 could be externalized under stimulation conditions. Concurrent CD39/CD73 expression is crucial since both molecules—co-expressed in either the same cells or the closing cells—may interact to integrate a receptor to generate extracellular adenosine (ultimately delimited by CD73 activity), which is endowed with extensive anti-inflammatory abilities [23,24]. 

CD39 and CD73 expression may be upregulated by inflammatory and anti-inflammatory modulators [25,26]. This study selected IFN-γ because it has been described as the best preconditioning ex vivo stimulus to improve the MSCs’ immunoregulatory properties [17]. This study showed that after 24 h of IFN-γ (1000 IU/mL) stimulation, the hGMSCs significantly upregulated surface CD39/CD73 coexpression and improved adenosine production. This effect was subject to CD39 because its expression was upregulated, while CD73 remained constitutive (Figure 3). These results are comparable to those obtained by Saldanha-Araujo et al. [16], who showed that MSCs cocultured with activated T cells or their supernatant increased CD39/CD73 coexpression as well as adenosine production and that it was dependent on CD39 upregulation. 

Although controversial data from in vivo studies have claimed that preconditioning MSCs with inflammatory cytokines is not essential for improvement in function in some inflammatory conditions [27], additional reports support the beneficial effect of the inflammatory stimulation of MSCs with IFN-γ [15,16,17], suggesting that the effect’s response could rely on different aspects, such as the kinds of cytokine, its concentration, and the kinds of cells. To our knowledge, no study has explicitly examined the in vitro effect of IFN-γ preconditioned hGMSCs on PBMC activation and maturation; therefore, this study first applied this strategy to evaluate and validate this effect. The results showed that compared to hGMSCs, hGMSC-γ exhibited better PBL proliferation inhibition of around two-fold (Figure 4A,B). In addition, while looking for a likely tolerogenic mechanism, this study examined whether hGMSCs could induce regulatory T cell expansion by scrutinizing CD39 expression because this marker has been widely linked to blood regulatory T cells [28,29]. The analysis revealed that CD39 was upregulated after MLR, and its expression remained higher on PBL when cocultured with both hGMSCs and hGMSC-γ, being significantly superior in PBL cocultured with hGMSC-γ (Figure 4). 

IFN preconditioning allowed the hGMSCs to sustain their immunoregulatory capability, while avoiding being affected by an inflammatory stimulus such as LPS, since PBMC activation in the presence of hGMSC-γ showed a statistically better reduction of CD40 and CD83 expression (in both frequency and MFI) (Figure 5); meanwhile, PBMC activation in the presence of not-IFN-γ preconditioned hGMSCs displayed low CD40 reduction and even CD83 expression was increased. This effect might be due to TLR-4 expression in MSCs, wherein LPS recognition might affect their regulatory functions during stimulation [30,31], supporting the evidence that BMMSCs negatively affects the expression of maturation and co-stimulation markers like CD83, CD40, CD80, or CD86 in dendritic cells stimulated with LPS [32,33]. However, hGMSCs primed with the IFN-γ’s effect on PBMCs have not yet been analyzed. 

The analysis of the impact of hGMSC-γ on pDCs in this study included CD40 expression (activation) and IFN-α secretion (functionality) as essential immunogenic pDCs-related factors induced by TLR-7 ligation (R848) [34]. Both hGMSCs and hGMSC-γ significantly regulated pDC activation and functionality, but, interestingly, such an effect was improved in hGMSC-γ (Figure 6). Although several studies have evidenced the efficacious immunoregulatory effects of MSCs on different immune cells, this study first demonstrated the hGMSCs’ direct immunoregulation on peripheral human blood pDCs. A recent study showed that a human umbilical cord blood-derived MSC infusion in a psoriasis mouse model reduced the disease severity by downregulating IFN-I production. However, the authors did not analyze the direct effect of the MSCs on the pDCs; instead, they explored the impact of coculturing splenic pDCs with a neutrophil supernatant derived from MSC-treated mice with psoriasis [35].

Aiming to strengthen the likely inhibitory contribution of the “purinergic halo” [16], this study additionally analyzed the hGMSC-derived effect on CD39 and CD73 expression on pDCs. pDCs activation significantly upregulated CD39/CD73 coexpression; hGMSCs increased this effect and improved it when hGMSC-γ was present (Figure 7). A previous investigation from our research group demonstrated that human peripheral blood pDCs possess constitutive extracellular CD39 expression, while CD73 is localized intracellularly. Both ectoenzymes are upregulated, and adenosine production is significantly augmented after pDC activation. Likewise, when CD73 activity was neutralized in pDC, these cells showed increased allostimulatory capability on T cells [36].

Thus, the results of the current study show that hGMSCs regulate pDC activation via adenosine, and this effect might be reinforced by the “purinergic halo” expressed between neighboring hGMSCs and pDCs. This means that when pDCs were stimulated in the presence of CD73 activity-neutralized hGMSC-γ (inhibition of adenosine production), this restored the effect promoted by the INF-γ preconditioning since CD40 expression and IFNα levels in pDCs were similarly inhibited by hGMSCs (Figure 6 and Figure 7). In line with our results, some studies have shown that neutralizing CD39 or CD73 activity alters the immunoregulatory effect of different MSCs, including hGMSCs [13,37,38], supporting the relevance of these ectoenzymes to produce adenosine and mediate the regulatory functions of MSCs. Therefore, this study provides new evidence about the potential of hGMSCs for designing further pre-clinical research on treating pDC-related chronic inflammatory disease.

## 5. Conclusions

Primary cultured hGMSCs regulate the phenotype, activation, and functionality of human blood pDCs. IFN-γ preconditioning improves the hGMSCs’ immunoregulatory abilities regarding pDCs via adenosine. Given the increase in CD39/CD73 expression and adenosine production after hGMSC and pDC stimulation, the “purinergic halo” between these cells might reinforce the hGMSCs’ inhibitory effect.

## Figures and Tables

**Figure 1 biomolecules-14-00658-f001:**
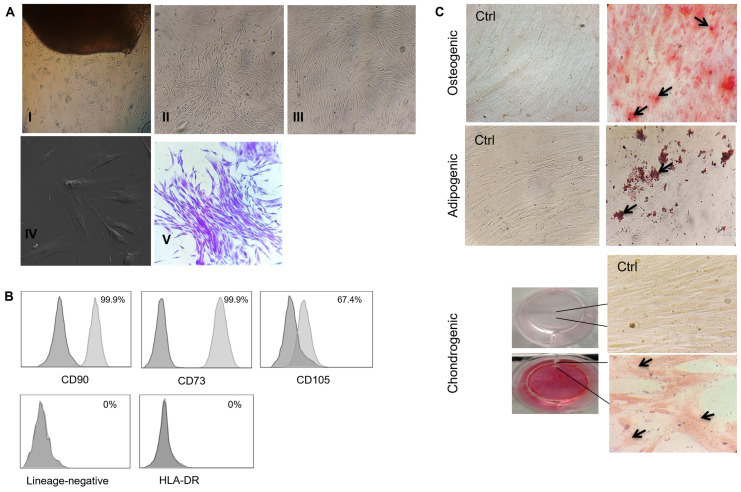
Isolation and characterization of human gingival tissue-derived MSCs. (**A**) Initial culture of hGMSCs: I, the migration of elongated adherent cells close to the tissue after around 5–7 days of culture; II and III, expansion of proliferating cells at 20–25 days of culture; IV, phase contrast microscopy of elongated, flat, spindle cells; V, CFU-F assay. (**B**) Phenotype analysis of hGMSCs: (dark gray histogram = isotype control; light gray histogram = specific antibody). High frequency of CD90+, CD73+, CD105+, and negative frequency of HLA-DR and hematopoietic lineage-negative (CD34, CD45, CD14, CD19). (**C**) Multipotent differentiation capacity: upper, Alizarin Red Oil staining reveals osteogenic differentiation (straight lines show the area of 40× amplification, and the arrow points to the mineralized deposition of calcium); Middle, Oil-Red O staining reveals adipogenic differentiation (the arrows point to the formation of lipids drops in the cytoplasm); lower, Safranin O staining reveals the formation of an extracellular matrix fibril in chondrogenic differentiation (the arrow points to the proteoglycan globular structure). Images are representative of 6 independent experiments.

**Figure 2 biomolecules-14-00658-f002:**
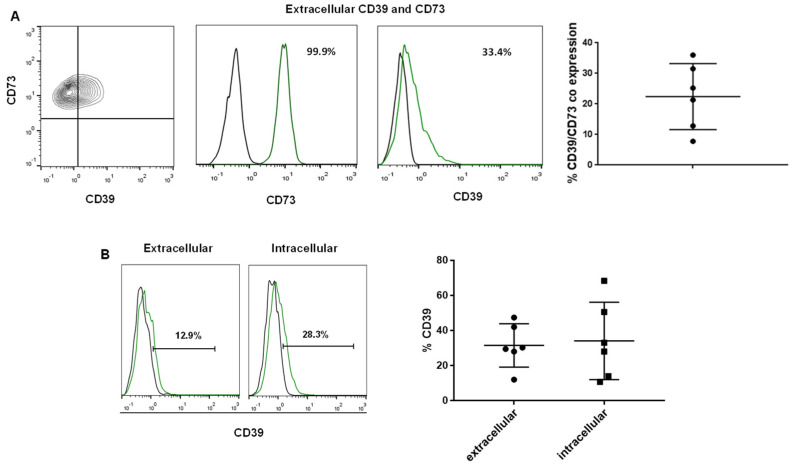
Analysis of CD39 and CD73 expression on hGMSCs. (**A**) Contour plot showing CD39 and CD73 coexpression and histograms showing the individual CD73 or CD39 extracellular expression (black line = isotype control and green line = specific antibody); the graphic shows the frequency of CD39/CD73 cells. (**B**) Histograms show the extracellular (**left**) and intracellular (**right**) expression of CD39; each black dot in the graphics displays the frequency of CD39+ cells per individual of *n* = 6.

**Figure 3 biomolecules-14-00658-f003:**
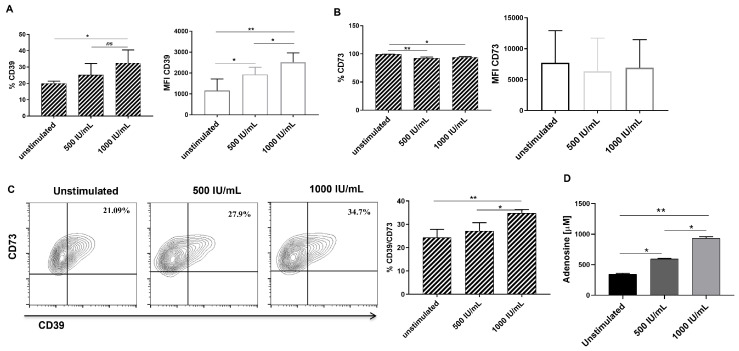
Analysis of IFN-γ stimulation on the members of the purinergic pathway on hGMSCs. Cells were stimulated with 500 or 1000 IU/mL IFN-γ for 24 h. (**A**,**B**) Graphics of frequency of positive cells (%) and expression (MFI) of CD39 and CD73, respectively. (**C**) Representative contour plots and graphics showing CD39/CD73 double-positive cell frequency. (**D**) Extracellular adenosine production by hGMSCs after IFN-γ stimulation. *n* = 6; the Kruskal–Wallis test was performed between each condition, where * *p* < 0.05, ** *p* < 0.01.

**Figure 4 biomolecules-14-00658-f004:**
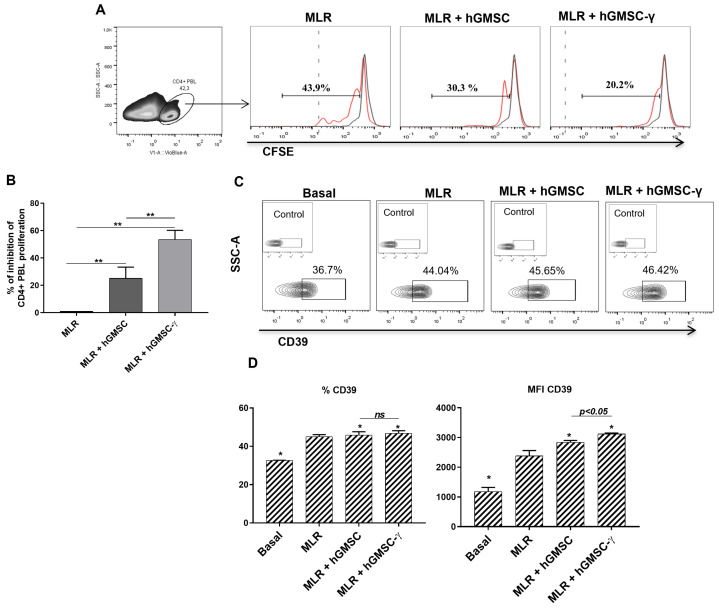
Validation of primary cultured hGMSCs’ immunoregulatory properties and evaluation of the IFN-γ-preconditioning effect. A mixed leukocyte reaction was performed in the absence (MLR) or the presence of IFN-γ-preconditioned (MLR + hGMSC-γ) or not-preconditioned hGMSCs (MLR + hGMSC). (**A**) Density plot of CD4+ PBL, gated to analyze CFSE dilution. Black histograms show the CFSE basal staining, while red histograms display the percentage of CFSE dilution (proliferation) after the coculture. (**B**) Graphic shows the inhibition percentage of CD4+ PBL proliferation concerning the MLR. (**C**) Contour plot of CD39+ PBL frequency. (**D**) Graphic of CD39+ PBL frequency and CD39 expression (MFI). The Mann–Whitney test was performed for statistical analysis (*p* < 0.05) of the results obtained in three independent duplicated experiments. The statistical difference (*) is displayed as the difference between each of the MLR conditions (* *p* ≤ 0.05 and ** *p* ≤ 0.01). Control: isotype control.

**Figure 5 biomolecules-14-00658-f005:**
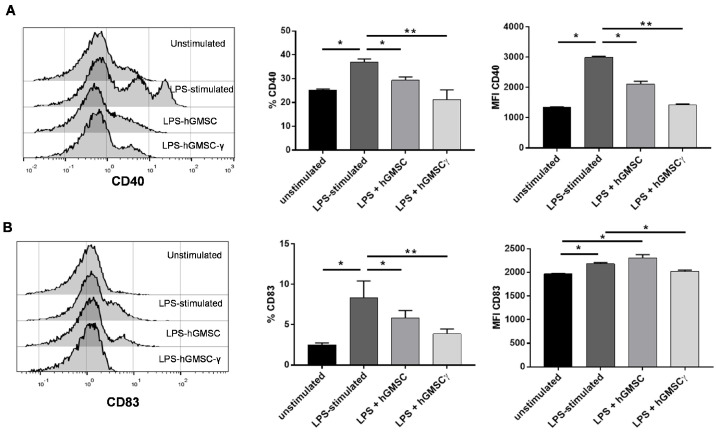
Validation of hGMSCs’ immunomodulatory capacities and IFN-γ preconditioning on LPS-stimulated PBMC. PBMCs were stimulated with LPS in the absence (LPS-stimulated) or in the presence of IFN-γ preconditioned (LPS + hGMSC-γ) or not-preconditioned hGMSCs (LPS + hGMSCs). Histograms and graphics show the frequency (%) and expression (MFI) in terms of CD40 (**A**) and CD83 (**B**). Histograms are representative of duplicates of 3 independent experiments. A Mann–Whitney test was performed for statistical analysis. The statistical difference (* *p* ≤ 0.05 and ** *p* ≤ 0.01) is displayed as the difference between each condition and the LPS-stimulated PBMCs.

**Figure 6 biomolecules-14-00658-f006:**
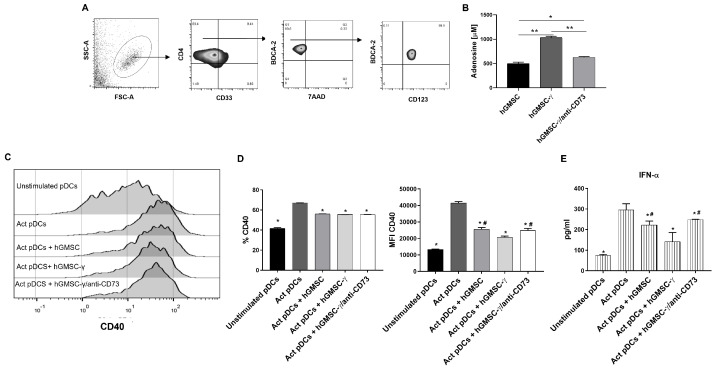
Analysis of the immunomodulatory effect of hGMSC-γ on pDC and the contribution of the purinergic pathway. The pDCs were stimulated with R848 for 24 h in the absence (Act pDCs) or the presence of hGMSCs (Act + hGMSC) or IFN-γ preconditioned hGMSCs (Act pDCs + hGMSC-γ), or IFN-γ preconditioned and anti-CD73-pretreated hGMSCs (Act pDCs+ hGMSC- γ/anti-CD73). (**A**) Gating strategy to evaluate the viability and purity of 7AAD negative, BDCA-2+CD123+CD4+CD33-blood-purified pDC. (**B**) Adenosine quantification, as described in Section 2. (**C**) Representative histograms showing CD40 expression. (**D**) Graphics showing the frequency of CD40+ and CD40 expression (MFI) in pDC. (**E**) IFN-α secretion in the supernatants in the cocultures. Data show an analysis of the results obtained from three independent experiments performed in duplicate. The Mann–Whitney test was performed for statistical analysis. * = *p* ≤ 0.05, ** = *p* ≤ 0.01 of statistical differences compared to Act pDCs; and # = *p* ≤ 0.05 of statistical differences compared to the Act pDCs + hGMSC-γ condition.

**Figure 7 biomolecules-14-00658-f007:**
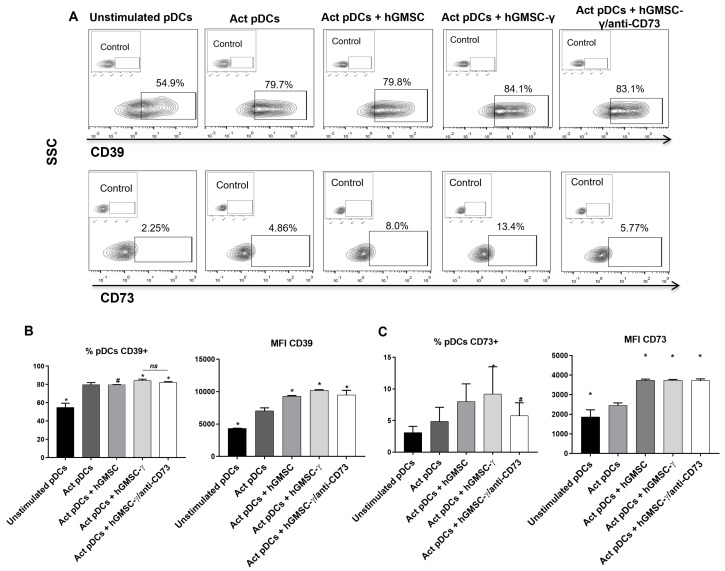
Analysis of hGMSCs’ influence on the expression of the purinergic halo in pDC. (**A**) Representative contour plots showing CD39 and CD73 expression on pDCs. (**B**,**C**) Graphics of the statistical analysis of frequency (%) and expression (MFI) of CD39 and CD73 on pDCs, showing the results obtained in three independent experiments made in duplicate. The Mann–Whitney test was performed for statistical analysis. * = *p* ≤ 0.05 of statistical differences compared to Act pDCs; and # = *p* ≤ 0.05 of statistical differences compared to the Act pDCs + hGMSC-γcondition. Control: isotype control.

## Data Availability

Data are contained within the article.

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
