# Peer review of "IFN-γ-Preconditioned Human Gingival-Derived Mesenchymal Stromal Cells Inhibit Plasmacytoid Dendritic Cells via Adenosine"

_biomolecules, 2024, doi:10.3390/biom14060658_

Round 1

Reviewer 1 Report

Comments and Suggestions for Authors

The authors studied the immunomodulatory function of IFN-γ-preconditioned human gingival-derived mesenchymal stromal cells (hGMSCs) in CD39/CD73 expression, adenosine production, and pDC activation. Generally, the authors provided data supporting the hypothesis that IFN-γ strengthened the immunomodulatory functions of pGMSCs on pDCs.

Below are a few comments that need attention:

1.     The markings and texts on the figures are too small to read, especially in Figures 3 and 6.

2.     In Figure 6B, a control group showing the adenosine production by the pDCs without MSCs is required.

3.     The authors mainly applied non-parameteric statistics to analyze the data. Accordingly, it is more appropriate to present the data with median and interquartile range (IQR), instead of mean with standard deviation.

Author Response

Reviewer 1

Comments and Suggestions for Authors

The authors studied the immunomodulatory function of IFN-γ-preconditioned human gingival-derived mesenchymal stromal cells (hGMSCs) in CD39/CD73 expression, adenosine production, and pDC activation. Generally, the authors provided data supporting the hypothesis that IFN-γ strengthened the immunomodulatory functions of pGMSCs on pDCs.

Below are a few comments that need attention:

  1. The markings and texts on the figures are too small to read, especially in Figures 3 and 6.

Answer: Following up on the reviewer´s comment, the authors improved the quality of the marking and text in each picture.

  1. In Figure 6B, a control group showing the adenosine production by the pDCs without MSCs is required.

Answer: The reviewer´s interorgan about the adenosine production by pDCs is pertinent. Hence, our research group has already published this information (PMID: 37055675). Because the present study focuses on analyzing the impact of IFN-γ to strengthen the immunomodulatory abilities of hGMSC and their effect on the component of the purinergic halo on these cells, Figure 6B displays the production of adenosine by hGMSC without IFN-γ stimulation, IFN-γ preconditioned, and with an anti-CD73 neutralizing antibody.

  1. The authors mainly applied non-parameteric statistics to analyze the data. Accordingly, it is more appropriate to present the data with median and interquartile range (IQR), instead of mean with standard deviation.

Answer: The authors appreciate the comments delivered by the reviewer. All graphics have been modified.

Reviewer 2 Report

Comments and Suggestions for Authors

Overall, this paper focuses on the use of human gingival derived MSCs as an alternative to the gold standard, but difficult to isolate, bone marrow MSCs to inhibit plasmacytoid dendritic cells. The manuscript is well written, and the objectives, methods and results are well addressed, congratulations to the authors. However, the manuscript would benefit from some minor revisions:

·      Please, ensure consistency in the term MSCs, in the abstracts it incorrectly states that it is the abbreviation for mesenchymal stem cells, while the title and introduction use the term stromal cells.

·      While MSC licensing or preconditioning has been proven to improve the immunoregulatory properties of MSCs, it has also been described to be detrimental in certain disease settings like psoriasis (PMID: 37373278). This should be included in the intro or discussion.

·      To satisfy ISCT criteria for MSC characterization, 98% of the cells should be CD73, CD105 and CD90 positive while being negative (<2% positive) for the other markers. Please quantify the expression of these markers since it looks like there is no clear separation between isotype control and stained. Note that culture with FBS can reduce the expression of CD105.

·      Throughout the results section, do not just mention that IFN treatment increases the expression of CD39, you also need to include the values with and without treatment and the p value in the text. This applies to all types of result description throughout the text, not just to this specific example.

·      In Figure 4C, include isotype control as negative control to ensure that the gating is appropriate. This also applies to Figure 7A and overall, to all your flow plots

·      Include that BM MSCs are not only the most investigated, but they are also considered the gold standard.

Author Response

Reviewer 2

Comments and Suggestions for Authors

Overall, this paper focuses on the use of human gingival derived MSCs as an alternative to the gold standard, but difficult to isolate, bone marrow MSCs to inhibit plasmacytoid dendritic cells. The manuscript is well written, and the objectives, methods and results are well addressed, congratulations to the authors. However, the manuscript would benefit from some minor revisions:

 Answer: The authors appreciate the reviewer's interest in evaluating the manuscript. All comments have been revised to improve the manuscript's quality.

  1. Please, ensure consistency in the term MSCs, in the abstracts it incorrectly states that it is the abbreviation for mesenchymal stem cells, while the title and introduction use the term stromal cells.

Answer: The authors thank the reviewer for noticing the inconsistency in MSC terminology. The error in the abstract section has been modified as “mesenchymal stromal cells”

  1. While MSC licensing or preconditioning has been proven to improve the immunoregulatory properties of MSCs, it has also been described to be detrimental in certain disease settings like psoriasis (PMID: 37373278). This should be included in the intro or discussion.

Answer: The authors appreciate the suggestion of the reviewer. Following up on the reviewer´s comment, the author added the former evidence to support the claim in the discussion section; the following statement was included: “ Although controversial data from in vivo studies have claimed that preconditioning of MSC with inflammatory cytokines is not essential for improvement function in some inflammatory condition [29], additional reports support the benefic effect of inflammatory stimulation of MSC with IFN- γ [15–17], suggesting that the effect´s response could rely on different aspects such as the kind of cytokine, concentration, kind of cells.”

  1. To satisfy ISCT criteria for MSC characterization, 98% of the cells should be CD73, CD105 and CD90 positive while being negative (<2% positive) for the other markers. Please quantify the expression of these markers since it looks like there is no clear separation between isotype control and stained. Note that culture with FBS can reduce the expression of CD105.

Answer: Following the reviewer’s recommendation, the different markers were quantified, and the figures were edited.

  1. Throughout the results section, do not just mention that IFN treatment increases the expression of CD39, you also need to include the values with and without treatment and the p value in the text.

Answer: The authors appreciate the reviewer´s pertinent suggestion to improve the quality of the manuscript. Values of expression and p value have been added for all types of results.

  1. In Figure 4C, include isotype control as negative control to ensure that the gating is appropriate. This also applies to Figure 7A and overall, to all your flow plots

Answer: Following the reviewer´s request, the isotype control was added.

  1. Include that BM MSCs are not only the most investigated, but they are also considered the gold standard.

Answer: Following the reviewer's recommendation, the following statement was included in the introduction: “Bone marrow-derived MSC is the most investigated and considered the “gold standard” for MSC research.”